# Unstimulated Parotid Saliva Is a Better Method for Blood Glucose Prediction

**Yangyang Cui** [1,2,3,†], **Hankun Zhang** [1,2,3,†], **Jia Zhu** [3], **Lu Peng** [1,3], **Zhili Duan** [1,3], **Tian Liu** [1,3], **Jiasheng Zuo** [1,3], **Lu Xing** [1,3], **Zhenhua Liao** [3], **Song Wang** [3,*] and **Weiqiang Liu** [1,2,3,*]

1   Tsinghua Shenzhen International Graduate School, Tsinghua University, Shenzhen 518055, China;
    cuiyy20@mails.tsinghua.edu.cn (Y.C.); zhanghk20@mails.tsinghua.edu.cn (H.Z.);
    pengl20@mails.tsinghua.edu.cn (L.P.); dzl20@mails.tsinghua.edu.cn (Z.D.);
    liut20@mails.tsinghua.edu.cn (T.L.); zjc20@mails.tsinghua.edu.cn (J.Z.); xingl20@mails.tsinghua.edu.cn (L.X.)
2   Department of Mechanical Engineering, Tsinghua University, Beijing 100084, China
3   Biomechanics and Biotechnology Lab, Research Institute of Tsinghua University in Shenzhen,
    Shenzhen 518057, China; zhuj@tsinghua-sz.org (J.Z.); liaozh@tsinghua-sz.org (Z.L.)
*   Correspondence: wangs@tsinghua-sz.org (S.W.); weiqliu@hotmail.com (W.L.);
    Tel.: +86-0755-265-586-33 (S.W.); +86-0755-265-513-76 (W.L.)
†   These authors contributed equally to this work.

**Abstract:** Objective: Saliva glucose has been widely used in diagnosing and monitoring diabetes, but the saliva collection method will affect saliva glucose concentration. So, this study aims to identify the ideal saliva collection method. Method: A total amount of six saliva collection methods were employed in 80 healthy participants in the morning. Besides, three unstimulated saliva methods were employed in another 30 healthy participants in the morning; in the meantime the blood glucose of these 30 participants was detected with a Roche blood glucose meter. The glucose oxidase method with 2, 4, 6-tribromo-3-hydroxybenzoic acid (TBHBA) as the chromogen has been improved to be suitable for healthy people, through the selection of the optimal pH value and ionic strength of the reaction system. This method was used for the detection of saliva glucose. Results: The improved method obtained absorbance at the wavelength of 520 nm, and the optimized parameter combination was pH 6.5 and 5 mg/dL NaCl. The lower limit of glucose detection was 0.1 mg/dL. Unstimulated saliva glucose concentration was higher than stimulated saliva glucose concentration. Unstimulated parotid saliva glucose concentration was the highest. Besides, unstimulated saliva glucose has a better normal distribution effect. Meantime, it was found that unstimulated parotid saliva was the most highly correlated with blood glucose ($R^2 = 0.707$). Conclusions: the saliva collection method was an important factor that affected saliva glucose concentration. Unstimulated parotid saliva was the most highly correlated with blood glucose, which provided a reference for prediction of diabetes mellitus.

**Keywords:** saliva; glucose; methods; diabetes mellitus; sample collection

## 1. Introduction

Diabetes mellitus (DM) is a globally common chronic disease affecting humans, which remains one of the major health concerns of the 21st century [1]. Without urgent and sufficient action, it is predicted that 578 million people will have DM in 2030 and the number will increase by 51% reaching 700 million in 2045 [2,3]. Blood glucose measurement is an indispensable method for screening and controlling DM. However, routine blood glucose detection requires invasive venipuncture or acupuncture, which brings pain to the patient and affects the patient's enthusiasm for blood glucose monitoring [4,5]. Therefore, non-invasive blood glucose monitoring has attracted great attention. Among the most non-invasive methods, saliva glucose, replacing blood glucose, has major significance in monitoring these conditions. This research area has already generated a plethora of previous scholarly work [6,7]. Caixeta et al. [8] showed that saliva was a promising solution

for the detection and monitoring of DM. Rodrigue et al. [9] pointed out that saliva, like blood, can reflect changes in human physiological functions, so it may be a substitute for early detection and monitoring of DM. Meanwhile, saliva collection is convenient, safe, non-invasive, with no risk of infection, and painless to patients. Therefore, people pay more and more attention to it in experimental research and clinical use [10,11].

Although saliva glucose reflects the health of the human body, its use as a diagnostic fluid has been hindered and neglected, mainly because of the lack of standardized saliva collection methods [12,13]. Most studies use saliva in diagnosis using different collection methods and often lack clear sampling processes [14,15]. This makes it difficult to compare the results of different studies [16]. In general, most studies view saliva incorrectly as a homogeneous body fluid. However, saliva is not a solitary fluid and cannot be viewed as such. Instead, it is a complex mixture consisting of the secretions of three main glands (parotid, submandibular, and sublingual), each of which secretes a characteristic type of saliva, along with hundreds of small salivary glands, gingival crevicular fluids and debris [17]. It is also unstable, but constantly changing, and its composition is affected by other factors such as sampling method, environment, oral hygiene, psychological status and general health [18]. Thus, it is necessary to establish precise standards for saliva collection [19], such as the type of saliva glucose, i.e., saliva produced by whole saliva or specific glands, and whether the sample was collected after stimulation [20,21]. Besides, most of the carbohydrates that are present in saliva are either synthesized in situ in the salivary glands and/or transported from blood capillaries into saliva by diffusion, active transport and/or ultra-filtration [22]. The glucose in saliva may also undergo modifications due to underlying pathological conditions and/or as a result of exposure to drugs and other compounds or solutions. Our understanding of the glucose present in saliva during a normal healthy physiological state, as opposed to a pathological condition, requires further investigation in order for saliva to become a sample of choice for diagnostic and treatment purposes.

Therefore, this study aims to identify the ideal saliva collection method. A total of six saliva collection methods were employed in 80 healthy participants in the morning. Besides, three unstimulated saliva methods were employed in another 30 healthy participants in the morning; in the meantime, the blood glucose of these 30 participants were detected with a Roche blood glucose meter. The glucose oxidase method with 2, 4, 6-tribromo-3-hydroxybenzoic acid (TBHBA) as the chromogen has been improved to be suitable for healthy people, through the selection of the optimal pH value and ionic strength of the reaction system. This method was used for the detection of saliva glucose.

## 2. Materials and Methods

### 2.1. Participants

In this study, 110 healthy participants with a mean $\pm$ SD age of 37.5 $\pm$ 4.1 years were included. 80 of these only collected saliva, and another 30 healthy participants not only collected saliva, but also had their blood glucose detected with a glucose meter (Roche Ltd., Basel, Switzerland). Inclusion criteria were good general health, age $\geq$ 18 years, and body mass index (BMI) $\leq$ 30 kg/m$^2$. All the participants were free of fever or cold and maintained exceptional oral hygiene on the day of collection. If oral examination indicated poor oral hygiene, hyposalivation, oral complaints, or other oral diseases (e.g., mucosal lesions, clinical signs of ongoing periodontal diseases), they were directly excluded from further involvement in the study. All participants signed an informed consent form. The collection of human blood and saliva samples was approved by the local ethics committee at Tsinghua University.

### 2.2. Glucose Collection

Smoking, brushing teeth, and eating or drinking 30 min before collection were avoided. Then the mouth was rinsed with water before collection to remove food residues in the oral

cavity [23]. A salivette (Sarstedt, 51.5134) (including untreated swabs and swabs stimulated by citric acid) was used to collect saliva glucose, including six collection methods.

For each participant, samples of parotid, sublingual/submandibular, and whole saliva were collected with and without stimulation (as shown in Figure 1), respectively denoted as unstimulated whole saliva (UWS), stimulated whole saliva (SWS), stimulated parotid saliva (SPS), unstimulated parotid saliva (UPS), unstimulated sublingual/submandibular saliva (USS), and stimulated sublingual/submandibular saliva (SSS).

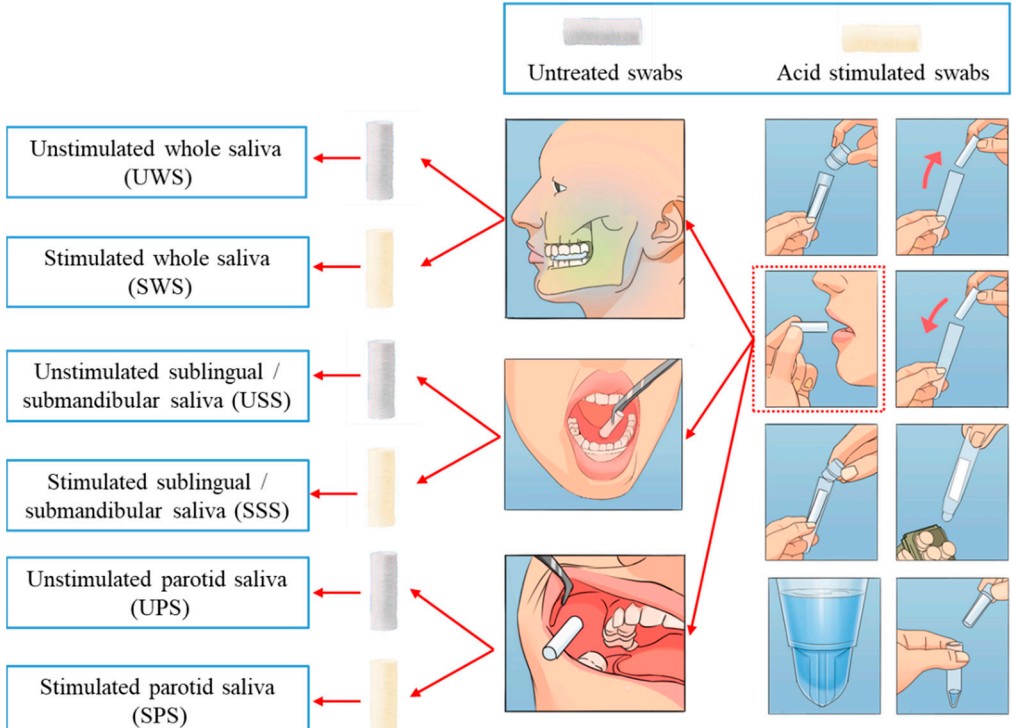

**Figure 1.** Six methods for collecting glucose; the swabs include untreated swabs and acid stimulated swabs, so the different swabs in parotid, sublingual/submandibular, and whole mouth represent unstimulated whole saliva (UWS), stimulated whole saliva (SWS), stimulated parotid saliva (SPS), unstimulated parotid saliva (UPS), unstimulated sublingual/submandibular saliva (USS), and stimulated sublingual/submandibular saliva (SSS).

All saliva glucose was collected in the same room in the morning. After completing the above steps, the cotton swab soaked with saliva was spit back into the collection tube, then the saliva was collected and weighed with an electronic balance (denoted as $Q_1$) after centrifugation, and was frozen directly at −20 °C for testing after collection [24]. The saliva flow rate (SFR) reflected the amount of saliva, calculated as in Equation (1).

$$\text{SFR} = \frac{Q_1 - Q_2}{T} \tag{1}$$

SFR was the saliva flow rate, $Q_1$ was the gross weight, $Q_2$ was the weight of collection tube, and $T$ was the collecting time.

### 2.3. Saliva Glucose Assay

The optimal pH and ionic strength of the reaction system were screened, and the glucose oxidase method using TBHBA as the chromogen was improved. Accurately, 20 mg of glucose was weighed before dissolving in artificial saliva (Phygene, pH = 7, Fuzhou, China), which was more than 99% similar to the saliva secreted by the real human body, then was transferred to a 100 mL volumetric flask, which was diluted to the mark to

obtain a 20 mg/dL artificial saliva glucose standard solution. Finally, it was diluted to 30 concentrations between 0.1–6 mg/dL, which were used for calibration of artificial saliva glucose solution measurements. Meanwhile, 0.25 mkat/LGOD and 0.17 mkat/LPOD as 1 mg/mL stock solutions according to the product instructions were prepared, and they were stored in aliquots and frozen at −20 °C. An appropriate amount was taken for each experiment and diluted to the required concentration with PBS. 4-aminoantibiotic Bilene (0.5 mmol/L) and TBHBA (5 g/L) were prepared by dissolving in PBS.

An ultra-micro ultraviolet spectrophotometer (Nano-Drop One Microvolume, Thermo Fisher Scientific, America) was used to obtain the absorbance spectrum of the sample. The wavelengths were selected at 500, 505, 510, 515, 520, 525, 530, 535 nm to obtain the absorbance spectrum of the sample, and the most relevant wavelength was selected. PBS with pH values of 5.6, 5.9, 6.2, 6.5, 6.8, 7.1, 7.4, 7.7, 8 were prepared. 0.1, 0.5, 0.9, 1.3, 1.7 and 2 mg/dL glucose sample were added to detection systems with different pH values, and the correlations were calculated. PBS (pH = 6.5) with NaCl concentration of 1, 3, 5, 7, 9, 11, 13, 15, 17 mg/dL were prepared. The sample group was set to 0.5, 1, and 1.5 mg/dL glucose sample, and the correlations were calculated.

The improved method was used to determine the linear range of glucose concentration detection using a newly constructed pH 6.5, NaCl concentration of 5 mg/dL reaction system to obtain the absorbance value of a glucose standard solution with concentrations of 0.1–6 mg/dL, the results obtained were drawn into a standard curve, and the method was used to measure the collected saliva samples.

Fasting blood glucose was tested before breakfast when all the saliva was collected using a glucose meter. Briefly, the index finger was disinfected with 70% alcohol, and a disposable sterile needle was used to obtain a drop of blood, which was collected on a glucose test strip and then inserted into the glucose meter. The blood glucose level was determined and recorded.

### 2.4. Statistics

SPSS was used to perform statistical analysis. The data were expressed as relative numbers, and $\chi^2$ was used for comparison between groups. The measurement data were conformed to the normal distribution and expressed as mean ± standard deviation ($\overline{x} \pm s$), and the *t*-test was used for comparison between groups. The Shapiro-Wilk test was used to test the normality of sample data. Non-normally distributed data were described in terms of minimum and maximum numbers, and normally distributed data were described in terms of ($\overline{x} \pm s$). Hypothesis testing would have insufficient sensitivity when the sample size was small, which would cause the results to lose use value, and if the data deviates slightly from normality the final test result would not have much impact, so box plots and qq plots could also be combined to perform statistical analysis. $p < 0.05$ indicated that the difference was statistically significant.

### 3. Results

#### 3.1. Saliva Detection Method

The reaction product of the improved methods had the highest absorbance value, measured at 510 and 520 nm wavelengths. However, at pH 6.5, the absorbance correlation was highest ($R^2 = 0.9948$) at 520 nm wavelength (as shown in Figure 2). Comprehensively, 520 nm of the maximum absorption wavelength and the pH 6.5 were chosen.

The correlations between different NaCl concentrations and absorbance when the glucose concentrations were 0.5 mg/dL, 1 mg/dL and 1.5 mg/dL are shown in Table 1. When NaCl were 5 mg/dL and 9 mg/dL, the correlations were the highest at 0.999, 0.999, but the Sy. x was the smallest at 5 mg/dL, and the correlation decreased when it was higher or lower than 5 mg/dL. Therefore, the optimal ion environmental concentration was selected as 5 mg/dL.

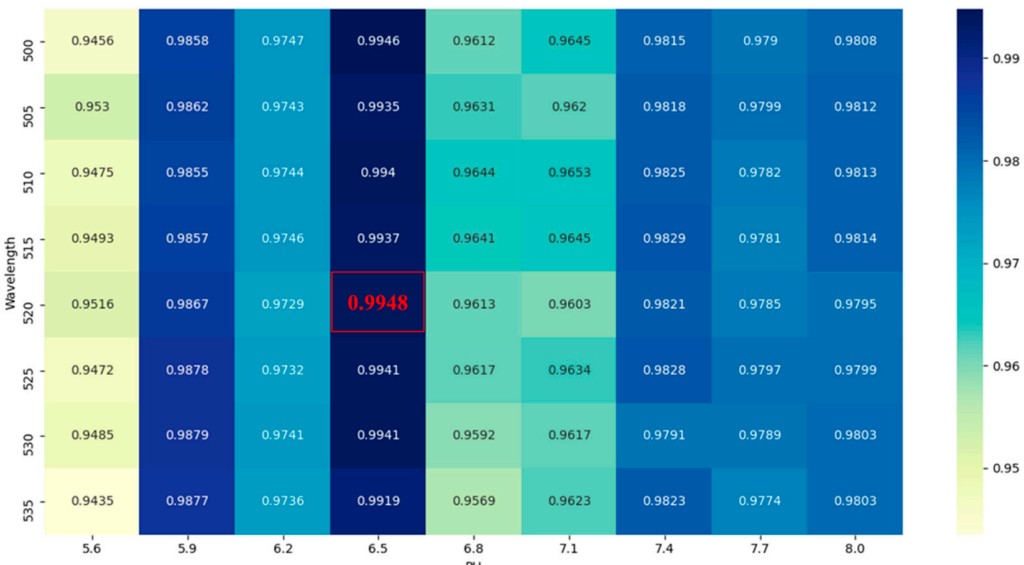

**Figure 2.** The influence of wavelength and pH on the absorbance of the sample. The red squared highlighting a value is the best result.

**Table 1.** Correlation between different NaCl concentration and absorbance with 520 nm wavelengths and pH 6.5.

| NaCl Concentration | Absorbance of Different Glucose Concentration | | | $R^2$ | Sy. x |
|---|---|---|---|---|---|
| | **0.5 mg/dL** | **1 mg/dL** | **1.5 mg/dL** | | |
| 1 mg/dL | $0.268 \pm 0.05$ | $0.572 \pm 0.15$ | $0.762 \pm 0.17$ | 0.9826 | 0.04654 |
| 3 mg/dL | $0.264 \pm 0.09$ | $0.474 \pm 0.13$ | $0.836 \pm 0.28$ | 0.977 | 0.06205 |
| 5 mg/dL | $0.28 \pm 0.09$ | $0.558 \pm 0.12$ | $0.826 \pm 0.27$ | 0.9999 ** | 0.004082 ** |
| 7 mg/dL | $0.3 \pm 0.05$ | $0.654 \pm 0.28$ | $0.75 \pm 0.22$ | 0.9012 | 0.1053 |
| 9 mg/dL | $0.238 \pm 0.03$ | $0.564 \pm 015$ | $0.902 \pm 0.23$ | 0.9999 * | 0.004899 * |
| 11 mg/dL | $0.262 \pm 0.15$ | $0.654 \pm 0.25$ | $1.302 \pm 0.61$ | 0.9802 | 0.1045 |
| 13 mg/dL | $0.25 \pm 0.15$ | $0.592 \pm 0.31$ | $0.992 \pm 0.45$ | 0.998 | 0.02368 |
| 15 mg/dL | $0.266 \pm 0.07$ | $0.504 \pm 0.21$ | $0.782 \pm 0.36$ | 0.998 | 0.01633 |
| 17 mg/dL | $0.368 \pm 0.19$ | $0.53 \pm 0.19$ | $0.872 \pm 0.24$ | 0.9592 | 0.07348 |
| 19 mg/dL | $0.254 \pm 0.21$ | $0.506 \pm 0.11$ | $0.744 \pm 0.33$ | 0.9997 | 0.005715 |

* Indicates the better result, ** Indicates the best result.

The absorbance was taken as the X axis and the glucose concentration as the Y axis, and linear regression was performed to obtain the standard curve equation as Y = 1.68X + 0.04, $R^2$ = 0.999, as shown in Figure 3. This showed that the improved method had a good linear relationship with the obtained absorbance value when the glucose concentration was in the range of 0.1–6 mg/dL. The lower limit of this range was 0.1 mg/dL, which fully met the sensitivity requirements for detecting the saliva glucose concentration of healthy people.

*3.2. Sample Characteristics*

Table 2 shows saliva glucose levels of the studied groups; the SFR in the unstimulated parotid saliva was the smallest, followed by the USS, and the largest was UWS. Stimulation of citric acid could also increase the SFR.

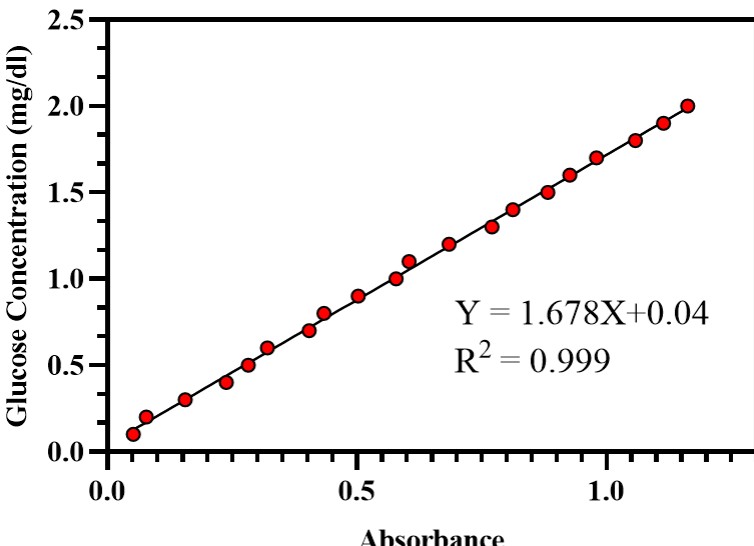

**Figure 3.** Standard curve of absorbance values of different concentrations of glucose.

**Table 2.** Saliva glucose levels of the studied groups.

| Collection Methods | UWS | SWS | UPS | SPS | USS | SSS |
|---|---|---|---|---|---|---|
| SFR (μL/min) | $1347 \pm 322$ | $1632 \pm 314$ | $113 \pm 21$ | $145 \pm 55$ | $413 \pm 89$ | $571 \pm 111$ |

SD: Standard Deviation, respectively denoted as unstimulated whole saliva (UWS), stimulated whole saliva (SWS), stimulated parotid saliva (SPS), unstimulated parotid saliva (UPS), unstimulated sublingual/submandibular saliva (USS), and stimulated sublingual/submandibular saliva (SSS).

Figure 4 shows absorbance of saliva glucose concentration at different saliva collection methods. It can be seen that the saliva concentration of unstimulated parotid saliva was significantly higher than the other five methods. At the same time, the stimulated saliva glucose concentration was significantly lower than the unstimulated saliva glucose concentration. In general, the amount of saliva collected by the stimulated method was much greater than that of the unstimulated method, but the stimulated saliva glucose concentration was lower than the unstimulated saliva glucose concentration.

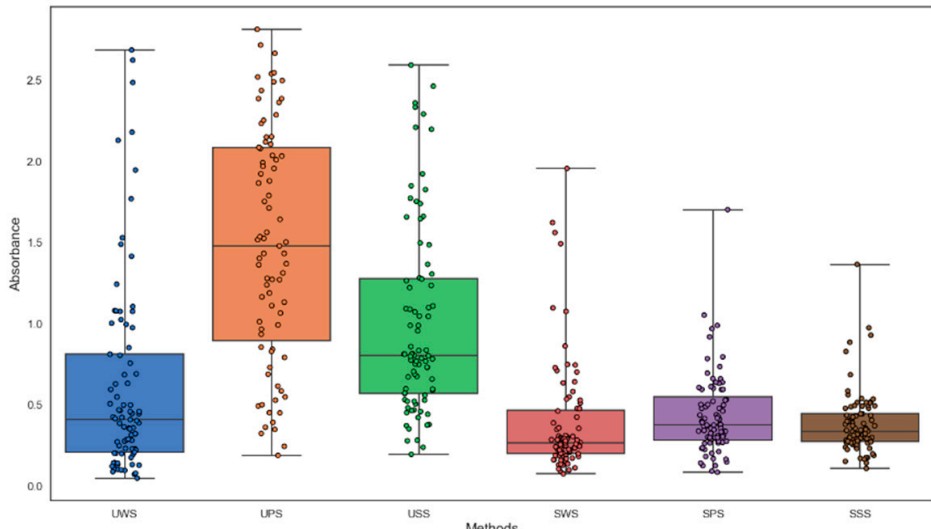

**Figure 4.** Absorbance of saliva glucose concentration at different saliva collection methods.

### 3.3. The Normal Distribution Curve of Each Collection Method

Figure 5 shows the normal distribution curve of different collection methods. The normal distribution curve reflected the distribution law of random variables, which indicated the potential of the data for saliva glucose testing. It can be seen that the unstimulated saliva methods had the better distribution law of random variables than the stimulated saliva methods. In general, unstimulated parotid saliva had better normal distribution than other saliva collection methods. So parotid glucose may be better used to respond to saliva glucose.

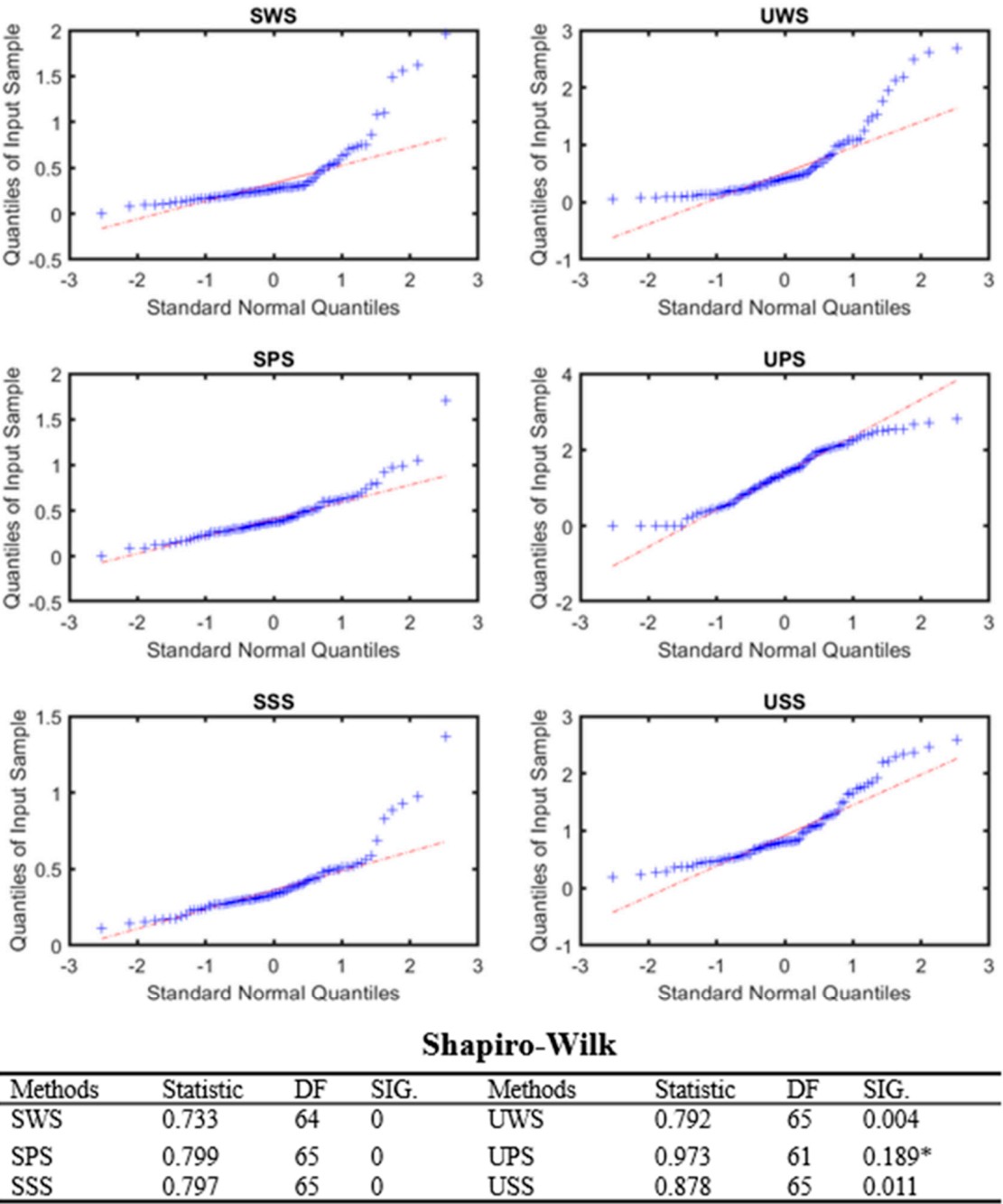

## Shapiro-Wilk

| Methods | Statistic | DF | SIG. | Methods | Statistic | DF | SIG. |
|---------|-----------|----|------|---------|-----------|----|------|
| SWS | 0.733 | 64 | 0 | UWS | 0.792 | 65 | 0.004 |
| SPS | 0.799 | 65 | 0 | UPS | 0.973 | 61 | 0.189* |
| SSS | 0.797 | 65 | 0 | USS | 0.878 | 65 | 0.011 |

**Figure 5.** The normal distribution curve of different collection methods. * Indicates that SIG is greater than 0.05 to accept the hypothesis. Respectively denoted as unstimulated whole saliva (UWS), stimulated whole saliva (SWS), stimulated parotid saliva (SPS), unstimulated parotid saliva (UPS), un-stimulated sublingual/submandibular saliva (USS), stimulated sublingual/submandibular saliva (SSS).

### 3.4. The Correlation of Blood Glucose and Unstiimulated Saliva Glucose

To assess the correlation of saliva glucose with human blood glucose levels, we performed a regression analysis. As shown in Figure 6, the linear regression equation for unstimulated parotid saliva glucose and blood glucose was $Y = 0.3435X + 4.671$, $R^2 = 0.7070$, $p < 0.0001$. The linear regression equation for unstimulated sublingual/submandibular saliva glucose and blood glucose was $Y = 0.4031X + 4.927$, $R^2 = 0.6211$, $p < 0.0001$. The linear regression equation for unstimulated whole saliva glucose and blood glucose was $Y = 0.4052X + 5.046$, $R^2 = 0.5114$, $p < 0.0001$.

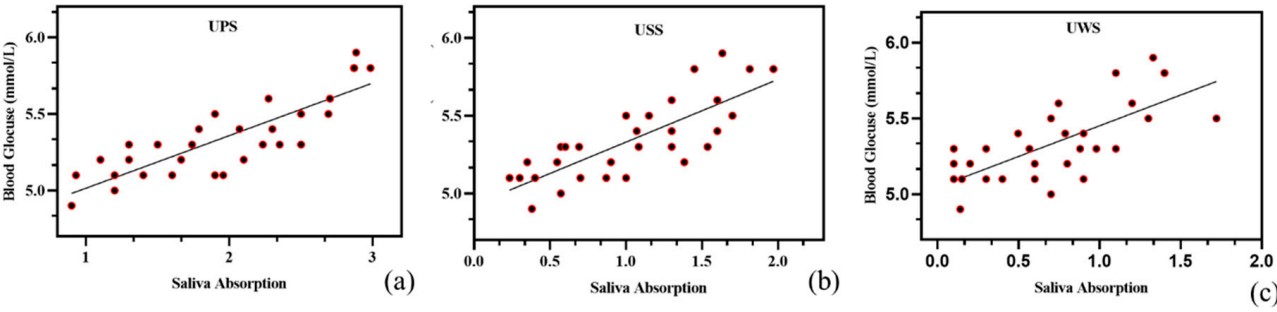

**Figure 6.** The correlation between saliva glucose and blood glucose: (**a**) unstimulated parotid saliva (UPS), (**b**) unstimulated sublingual/submandibular saliva (USS), (**c**) unstimulated whole saliva (UWS).

### 4. Discussion

Currently, non-invasive detection techniques based on saliva samples are basically targeted at DM patients. In this study, the glucose oxidase method with TBHBA as the chromogen has been improved to be suitable for healthy people. This method was used to compare the saliva glucose concentration of six different saliva collection methods for the first time.

This study found that the product has a maximum absorption at a wavelength of 520 nm, the optimal pH is 6.5, and the optimal NaCl concentration is 5 mg/dL. The environmental pH value will change or affect the dissociation state of the enzyme and the substrate to increase or decrease the enzyme activity. Therefore, the maximum activity of the enzyme requires the corresponding optimum pH value. The main catalytic enzymes in the test solution are GOD and POD. The activity of the former was in the range of pH 4.0 to 7.0, and the activity of the latter was in the range of pH 5.0 to 9.0. The optimal pH value of the reaction test solution measured in the experiment was 6.5, which was within the range of GOD activity and POD activity. Therefore, it was theoretically speculated that the pH value of 6.5 was the comprehensive optimal pH value of the same system where the two enzymes were located. Besides, it is found that its linear range, accuracy, and precision can meet the requirements of detection, and more importantly, makes the measurement process more standardized, reduces errors, and is simple and easy to implement.

Saliva, like plasma or serum, is a unique and complex body fluid. Sufficient saliva secretion is essential for maintaining oral health. The advantages of saliva assessment include the cost-effectiveness of non-invasive collection and screening of large populations [25]. Saliva is currently considered to be an excellent diagnostic biomarker for human characteristics [26]. In the present study, we found that the unstimulated saliva glucose levels were higher than in stimulated saliva, that unstimulated parotid saliva glucose level was higher than unstimulated sublingual/submandibular saliva, and that unstimulated whole saliva had the lowest level. The blood glucose and unstimulated parotid salivary glucose levels were significantly higher than levels in unstimulated sublingual/submandibular saliva and unstimulated whole saliva, and the glucose levels in parotid saliva were strongly correlated with blood glucose in healthy people. However, in another study, no correlation was found between saliva and plasma glucose levels [27]. Nonetheless, our study revealed a significant, strong correlation between parotid salivary and blood glucose levels but not between

mixed salivary and blood glucose levels, unstimulated sublingual/submandibular saliva and blood glucose; the glucose level in parotid saliva, but not that in unstimulated sublingual/submandibular saliva and unstimulated whole saliva, may thus reflect the blood glucose level. The different conclusions are mainly caused by different saliva collection methods. The six saliva collection methods in this study can make up for the shortcomings of existing research and provide the next step for the concentration of glucose in saliva. The determination of saliva has laid a good foundation and pointed out the direction for finding the most suitable method to collect saliva glucose.

The determination of saliva glucose concentration is a prerequisite for the development of saliva as a diagnostic and prognostic tool for DM biomarker discovery. In this case, it is important to keep the technical variability caused by sample collection and processing to a minimum so that inter-subject variability in health and disease states can be assessed reproducibly [28]. Single or mixed saliva can be collected. It should be noted that many unknown factors and unstable elements will affect the properties of mixed saliva. Saliva collected directly from a single gland is stable and not affected by oral conditions. Thus, it can accurately reflect blood glucose status. Saliva from the parotid gland is easily collected under unstimulated and stimulated conditions. Dhanya et al. [29] reported that, when saliva is collected under unstimulated conditions, the concentration of glucose in saliva is higher than under stimulated conditions, which is consistent with the conclusions obtained in this study. Other studies have found that there is no significant difference in the concentration of glucose in saliva collected under unstimulated and stimulated conditions [30], because participants may not be willing to accept acid stimulation, and the water concentration in stimulated saliva is higher. Besides, unstimulated saliva may be more representative of a normal physiological state. Takeda et al. [31] measured the saliva chemical concentration of healthy subjects under different conditions and found that, compared with stimulated saliva, almost all metabolites in unstimulated saliva were higher. Jha et al. [32] also found that, compared with stimulated saliva, the average saliva glucose level in unstimulated saliva of control and non-control DM patients was higher. Saliva collected directly from a single gland is stable and not affected by oral conditions. Therefore, it may accurately reflect blood glucose status. Moreover, as far as we know, this is the first study focused structurally on comparing the glucose expression of whole saliva and glandular saliva in a cohort of careful characterization and clinical examination. The results indicate that different collection methods provide significant differences in the snapshots of saliva glucose.

The limitation of our study is the relatively small sample size. Further studies with a larger sample size are necessary to confirm the correlation between blood glucose and saliva glucose, so as to design a saliva-based diagnostic test method for DM. In addition, there are still many problems in this study that need to be resolved and further explored. For example, the submandibular glands and sublingual glands are closely located, so it is difficult to separate saliva from these glands with certainty, which is why saliva is collected from both glands. How to distinguish sublingual saliva from submandibular saliva is also a direction that needs further research.

In summary, the results of this study indicate that different saliva collection methods provide significant differences in the snapshots of saliva glucose. Based on the comparison of unstimulated and stimulated saliva collection methods, it can be shown that, based on the simplicity and low variability of the collection method, UPS may be a preferred collection method. The results emphasize the importance of consistency when collecting saliva samples, which should be more important than the collection method itself.

## 5. Conclusions

In this study, the glucose oxidase method with TBHBA as the chromogen has been improved to become suitable for healthy people. The lower limit of the concentration range determined in this study was 0.1 mg/dL, which fully met the sensitivity requirements for detecting the concentration of saliva glucose in healthy people. The collection method was

an important factor that affected the saliva glucose concentration. This study demonstrated that parotid salivary glucose has potential as an indicator to monitor blood glucose.

**Author Contributions:** Conceptualization, Y.C.; methodology, H.Z.; software, J.Z. (Jia Zhu); validation, L.P.; formal analysis, Z.D.; investigation, T.L.; resources, L.X.; data curation, J.Z. (Jiasheng Zuo); writing—original draft preparation, Y.C.; writing—review and editing, H.Z.; visualization, H.Z.; supervision, S.W.; project administration, Z.L.; funding acquisition, W.L. All authors have read and agreed to the published version of the manuscript.

**Funding:** This project was supported by the Guangdong Basic and Applied Basic Research Foundation, the Innovation Commission of Science and Technology of Shenzhen Municipality and the Shenzhen Municipal Industrial and Information Technology Bureau.

**Institutional Review Board Statement:** The study was conducted according to the guidelines of the Declaration of Helsinki, and approved by the local ethics committee at Tsinghua University (protocol code 74 and date: 8 November 2021).

**Informed Consent Statement:** All study subjects signed an informed consent form, and the collection of human blood and saliva samples was approved by the local ethics committee at Tsinghua University.

**Data Availability Statement:** The study did not report any data.

**Acknowledgments:** This project was supported by the Guangdong Basic and Applied Basic Research Foundation (Grant No. 2020B1515120082), the Innovation Commission of Science and Technology of Shenzhen Municipality (Grant No. JCYJ20190807144001746, Grant No. JSGG20191129114422849) and the Shenzhen Municipal Industrial and Information Technology Bureau (Grant No. 20180309163834680).

**Conflicts of Interest:** The authors declare no conflict of interest.

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
