# Peer review of "Unstimulated Parotid Saliva Is a Better Method for Blood Glucose Prediction"

_applsci, doi:10.3390/app112311367_

Round 1
Reviewer 1 Report
Cui et al. report the comparison in glucose detection based on different saliva collection protocols. The work seems interesting and well addressed, however some considerations should be taken into account before publication.
During sample collection, how do the authors know that the saliva sample corresponds only to the area extracted and there is no cross contamination with saliva from other areas?
All the paper is based on the importance of glucose detection and monitoring. However this is critical in DM patients but not so critical or frequently needed in healthy patients. Can the authors discussed why they only considered healthy people for their trials and do not validate their conclusions with DM patients? Is there any limitation that should be highlighted on the method?
In figure 2 where the optimal conditions of pH and wavelength are shown there is a red squared highlighting a value that is not the correct one. Does it mean something? Is it an error?
I advice the authors to carefully read the manuscript and correct spelling mistakes and typos all along the text.
I also found the abbreviations used quite complex to follow and do not help the reader to identify and differentiate the samples according to the collection method.
In figure 3, given the accuracy of the plotted numbers the authors should include minor ticks that help to read the graph with higher accuracy.
In my opinion figure 4 is pretty complex to understand and read. The legend seem repetitive with squares and dot representing the same? I also think they need to include significance test in the plot to indicate which values are significant regarding the others.
The authors claim a trend according the saliva collection methods and glucose concentration. How do they know this is real, and there is no interference of cross contamination or artefacts arising from the lack of control on the kind of saliva collected.
In figure 5, I would advice to present only in the main text the main result, the data they really target and it makes the story complete and move the rest of plot to supporting information. It will make rather easier to read the figure and understand their point.
The authors in the discussion states that the limitations of the current saliva collecting methods are biased by the method itself but also the age, stimulation level or time or day, This makes me wonder then, how significant is then their final conclusion? How can they assume that validating this method for a reduced number of subjects and all of them within the same age and physical conditions is meaningful?

Reviewer 2 Report
Review on the manuscript «Comparing Glucose Concentration among Six Saliva Collection methods in the Newly Developed Saliva Glucose Detection System” submitted to Applied Sciences.
Minor comments.
- Section 2.3. The Nanodrop system does not “detect” wavelengths (as it is written), but allows to obtain the absorbance spectrum of the sample. Please correct.
- Section 3.1. PH to be corrected to pH.
- Figure 2 is difficult to understand. A 3D graph may be easier to read. In any case, this kind of date is expected to be provided as supplementary material, not in the core of the manuscript.
- Table 1 caption does not provide the wavelength used. Please correct.
- The conclusion is very unclear.
Major comments
- The method that the authors describe as “ultra-micro ultraviolet spectrophotometry” is simple spectrophotometry performed with a routine benchtop Nanodrop device. The authors should correct the manuscript accordingly.
- The chromogen/GOx mechanism should be detailed. It is not detailed at all in this version of the manuscript. These details will be used to explain why a linear behavior is expected as a function of the glucose concentration.
- Statistics are made from a very small population (20), which makes the conclusions disputable.
- Figure 6 should provide the time elapsed between the meal and the sampling.
- The authors claim that “an automated analysis method using TBHBA as the chromogen was developed for the detection of saliva glucose.” But nothing like that has been developed and described in this manuscript : the experiments are not automated, and the TBHBA/GOx couple has been developed for years, not by the authors.
- Many if not most references are not related to the topic of this manuscript.
In conclusion, this manuscript describes colorimetric experiments performed using routine protocols. It shows large variations in glucose concentration within the volunteers panel, which is not expected for a test which is supposed to be reliable and indicative of a DM. In addition, there are many studies dedicated to glucose concentration in saliva and its comparison with blood concentration, and this manuscript does no bring novel results to that matter.
Therefore, the manuscript should be rewritten to be re-focused on the real novelties, or submitted to a more specialized journal.
Round 2
Reviewer 2 Report
All major comments have been considered, so that the manuscript can now be published.